# Phenotypic memory in quorum sensing

**Ghazaleh Ostovar**[1], **James Q. Boedicker**[1,2] *

**1** Department of Physics and Astronomy, University of Southern California, Los Angeles, California, United States of America, **2** Department of Biological Sciences, University of Southern California, Los Angeles, California, United States of America

* boedicke@usc.edu

**Data Availability Statement:** All relevant data are within the manuscript and its Supporting Information files.

**Funding:** This work was supported by funding from NSF grant No. PHY-1753268 (JQB) and the Army Research Office MURI Award No.

## Abstract

Quorum sensing (QS) is a regulatory mechanism used by bacteria to coordinate group behavior in response to high cell densities. During QS, cells monitor the concentration of external signals, known as autoinducers, as a proxy for cell density. QS often involves positive feedback loops, leading to the upregulation of genes associated with QS signal production and detection. This results in distinct steady-state concentrations of QS-related molecules in QS-ON and QS-OFF states. Due to the slow decay rates of biomolecules such as proteins, even after removal of the initial stimuli, cells can retain elevated levels of QS-associated biomolecules for extended periods of time. This persistence of biomolecules after the removal of the initial stimuli has the potential to impact the response to future stimuli, indicating a memory of past exposure. This phenomenon, which is a consequence of the carry-over of biomolecules rather than genetic inheritance, is known as "phenotypic" memory. This theoretical study aims to investigate the presence of phenotypic memory in QS and the conditions that influence this memory. Numerical simulations based on ordinary differential equations and analytical modeling were used to study gene expression in response to sudden changes in cell density and extracellular signal concentrations. The model examined the effect of various cellular parameters on the strength of QS memory and the impact on gene regulatory dynamics. The findings revealed that QS memory has a transient effect on the expression of QS-responsive genes. These consequences of QS memory depend strongly on how cell density was perturbed, as well as various cellular parameters, including the Fold Change in the expression of QS-regulated genes, the autoinducer synthesis rate, the autoinducer threshold required for activation, and the cell growth rate.

## Author summary

Bacteria use a mechanism known as quorum sensing (QS) to collaborate when their numbers are high. In various QS systems, cells detect specific signals that trigger certain genes, resulting in increased production of certain molecules in response to these signals. Interestingly, these molecules can linger even after the initial signal is gone, which can resemble a form of "memory."

W911NF1910269 (JQB). The funders had no role in study design, data collection and analysis, decision to publish, or preparation of the manuscript.

**Competing interests:** The authors have declared that no competing interests exist.

Our theoretical study focuses on exploring this memory and the factors that influence it. To do this, we used simulations and models to examine how history of exposure to signals can affect the future response, when signals are removed, and cell density is reduced. We found that the prior exposure to signals can influence how bacteria respond in the future, but this effect occurs under specific conditions. This research contributes to our understanding of quorum sensing and how bacteria adapt to environmental changes.

## Introduction

Quorum sensing in bacteria is known as a mechanism to monitor cell density and coordinate gene expression within populations of cells [1–3]. QS relies on the synthesis and detection of diffusible signaling molecules, also known as autoinducers, by individual cells. As cell density increases, the concentration of released signals in the environment also increases. As the amount of external signal grows to a high concentration, the expression of target genes under QS regulation is initiated, exhibiting a switch-like behavior [4,5]. Since the concentration of signals in the environment is proportional to the population size, QS is considered a proxy for cell density. Several genes involved in collective behavior are subject to regulation under QS, including those responsible for virulence, bioluminescence, and biofilm formation [6]. Activation of QS in response to high signal concentrations leads to a transition of the intracellular concentrations of expressed target genes from a low to high state, referred to as the QS OFF and ON steady states, respectively [7,8]. Such switch-like behavior is often attributed to the presence of positive feedback loops within many QS circuits [5,9]. Positive feedback in QS is due to increased synthesis of proteins involved in signal transduction, the synthase and receptor proteins, upon activation. As a result, low and high signal concentrations as environmental inputs can induce two distinct phenotypes in a QS system, referred to as OFF and ON phenotypes, respectively. However, it remains unclear whether the QS response is solely determined by the current cell density and signal concentrations or is influenced by past exposure to signal.

Prior studies have shown that bacterial responses to current conditions may be influenced by past environmental conditions. Examples of such history-dependent behavior have been previously studied in metabolic systems, stress responses, and biofilm formation [10–13]. The *lac* operon in *E. coli* exemplifies history-dependent behavior in bacteria [11]. When transitioning from glucose to lactose as the carbon source, cells previously exposed to lactose exhibited shorter growth lag phases, indicating a memory of past lactose exposure. Overexpression of LacZ proteins further reduced the lag phase, suggesting that the memory effect is associated with carry-over of intracellular LacZ proteins, in which the duration of the lag phase reflected the strength of history-dependent behavior. Such a memory effect, resulting from the persistence of cellular components rather than genetic inheritance, is known as "phenotypic" memory [14,15]. Another instance of phenotypic memory in bacteria can be observed in the stress response of *Bacillus subtilis*. Mutlu et al. demonstrated that the ability of spores to recover is regulated by molecules transmitted from the progenitor cells to the spores during sporulation timing [13]. Moreover, previous exposure of *Pseudomonas aeruginosa* [12] cells to a surface imprints a memory of the surface through cyclic adenosine monophosphate (cAMP), enabling cells to gradually become better adapted for sensing and attachment upon returning to the surface. This phenotypic memory has been shown to potentially provide adaptive advantages in fluctuating environments [12].

Previous studies have also analyzed memory within QS. It has been observed that activation of competence in *Streptococcus pneumoniae* cells depended on the pH of media in past cultures [16]. Sappington et al. [17] demonstrated that LasR, a transcriptional activator involved in quorum sensing, can maintain its functional state even in the absence of the signaling molecule for a certain period. The study revealed that the presence of a reservoir of signal-free LasR within the cells enables the activation of transcription upon re-exposure to the signaling molecule. Another study demonstrated that overexpression of LuxR in *E. coli* cells carrying the Lux-I-LuxR quorum sensing circuit significantly shortened the onset timing of signaling supporting the effect of elevated receptor concentration is QS activation [17]. Moreover it has been observed that cells with previous QS activity can expedite activation in neighboring cells, suggesting the presence of a potential memory of prior stimulation in form of higher than basal rate of autoinducer synthesis [18].

In this study we aim to further investigate the phenomenon of phenotypic memory in QS. Specifically, we hypothesize that when bacterial cells transition from high to low cell densities, the presence of excessive receptor and synthase proteins carried over from the QS ON state may influence QS reactivation dynamics, thereby encoding a memory of past exposure. We utilize a mathematical model of QS to examine the consequences of biomolecule carry-over when cells move from a region of high cell density in QS ON state, to a region of low cell density. During this transition, molecular concentrations are expected to gradually transition back to levels associated with the OFF state. Our study aims to assess the potential carry-over effects and their impact on the expression of QS-regulated genes. To investigate the potential sources of phenotypic memory in QS, we evaluate the contribution of multiple components [19,20]. Considering that each biomolecule has a different abundance and decays at a different rate, it is plausible that multiple sources of memory may exist [10,21,22]. Furthermore, we develop a simplified analytical model to determine the parameters that control the strength and duration of phenotypic memory in quorum sensing.

## Methods

LuxR-LuxI regulatory circuit can be modeled through a set of ordinary differential equations (ODEs) [4] based on the mass-action kinetics formalism [23]. Such simplified models assume the concentration of biomolecules transformed by the reactions depend solely on the current amount of species, the rates at which these reactions proceed, and the stoichiometry of the reactions [4]. These equations were used previously to model QS dynamics [4,24]. The set of differential equations and the description of associated model parameters can be found in Eqs 1–8 and Table 1. These equations describe the dynamics of the quorum sensing machinery including receptor protein LuxR in unbound ($R$), monomeric ($RA$), and dimeric/complex forms ($C$), the synthase protein LuxI ($I$), AHL signaling molecule ($A$), a quorum sensing target gene ($G$), and cell number ($N$). Although the model should apply to many quorum sensing systems, it is inspired by the LuxR_LuxI circuit of *Vibrio fischeri*, a well-studied marine bacterium known for its bioluminescent interaction with the Hawaiian bobtail squid. This choice is based on the extensive availability of quantitative data regarding the QS system of *Vibrio fischeri*.

Briefly, quorum sensing regulated proteins (LuxR, LuxI, and the target QS-regulated protein) are produced at either a basal rate, when AHL concentration is low, or at an elevated rate, when high AHL concentrations results in receptor dimer formation. The protein concentration per cell is reduced due to both cell division and protein decay. The rate of protein degradation was set to reflect that cell division sets the rate that proteins are diluted during cell growth. To simplify the model, this rate of protein degradation does not change as cell growth slows near the saturating cell density. Separate degradation and production rates could have

**Table 1. Model parameters.**

| Parameter | Value | Description | Reference |
|---|---|---|---|
| $k_{-1}$ | $10[min^-1]$ | Dissociation rate of monomer (RA) | [4] |
| $k_{-2}$ | $1[min^-1]$ | Dissociation rate of dimer (C) | [4] |
| $k_{d1}$ | $100[nM]$ | Dissociation constant of monomer (RA) | [4, 27] |
| $k_{d2}$ | $20[nM]$ | Dissociation constant of dimer (C) | [4] |
| $k_1 = k_{-1}/kd1$ | $0.1[nM^-1min^-1]$ | Monomer (RA) formation rate | [4] |
| $k_2 = k_{-2}/kd2$ | $0.05[nM^-1min^-1]$ | Dimer (C) formation rate | [4] |
| b | $0.04[min^-1]$ | Synthesis rate of AHL by LuxI | [4] |
| D | $10[min^-1]$ | Diffusion rate of AHL through the cell membrane | [4, 28] |
| $k_3$ | $80 [nM^-1min^-1]$ | Activated expression rate of LuxI | estimated** |
| $k_4$ | $80 [nM^-1min^-1]$ | Activated expression rate of LuxR | estimated** |
| $k_5$ | $8 [nM^-1min^-1]$ | Basal expression rate of LuxI | estimated* |
| $k_6$ | $8 [nM^-1min^-1]$ | Basal expression rate of LuxR | estimated* |
| $k_{dR} = k_{dI}$ | $100 [nM]$ | Dimer (C) concentration at half maximum activity | estimated *** |
| $k_7$ | $0.006[min^-1]$ | Degradation rate of LuxI | [29] |
| $k_8$ | $0.15[min^-1]$ | Degradation rate of LuxR and monomer (RA) | [29] |
| $k_9$ | $0.005[min^-1]$ | Degradation rate of AHL | [30] |
| $k_{11}$ | $0.017[min^-1]$ | Growth rate | doubling time $\approx 40min$ |
| $k_{10}$ | $0.5[nM^-1min^-1]$ | Basal expression rate of QS target gene (G) | Estimated |
| r | $10^{-12}$ ml | $v_{cell}$ | [31] |

*Estimated based on [32], LuxI and LuxR are assumed to have approximately the same basal expression rates and QS activation at a cell density near $10^9$ cell/mL. ** Rate constants are estimated to achieve a Fold Change of 10 for LuxI ***Estimated to obtain half maximum activity at around 25–50 nM AHL concentration [33].

been implemented, dependent on the growth phase of the culture, however QS activation is expected to occur well before reaching density-dependent growth limitations. This simplification prevents a ramp up in protein levels during stationary phase. While dilution resulting from cell division is often the primary mechanism of protein degradation, it should be noted that LuxR is inherently unstable in its non-dimeric state, particularly when AHL concentrations are low [25], and thus is prone to rapid degradation. In this model, cells produce AHLs in proportion to the concentration of synthase protein present within each cell. The concentration of AHLs within the cells ($A$) and outside the cells ($A_{ex}$) is linked through diffusion across the cell membrane. Cell division in this model follows the logistic growth equation. Transcription and translation of mRNAs are not explicitly accounted for in the model, as these molecules have much shorter half-lives than the timescale of quorum sensing dynamics [26]. As a result, mRNAs are assumed to be in quasi-steady state and are not explicitly tracked within the model.

$$\frac{[dI]}{dt} = k_5 + \frac{k_3[C]}{k_{dI} + [C]} - k_7[I] - k_{11}[I] \qquad \text{Eq 1}$$

$$\frac{d[R]}{dt} = k_6 + \frac{k_4[C]}{k_{dR} + [C]} + k_{-1}[RA] - k_1[R][A] - k_8[R] - k_{11}[R] \qquad \text{Eq 2}$$

$$\frac{d[RA]}{dt} = 2k_{-2}[C] - 2k_2[RA][RA] - k_{-1}[RA] + k_1[R][A] - k_8[RA] - k_{11}[RA] \qquad \text{Eq 3}$$

$$\frac{d[C]}{dt} = k_2[RA][RA] - k_{-2}[C] - k_{11}[C] \tag{Eq 4}$$

$$\frac{d[A]}{dt} = b[I] - k_1[R][A] + k_{-1}[RA] - D([A] - [A_{ex}]) - k_9[A] - k_{11}[A] \tag{Eq 5}$$

$$\frac{d[A_{ex}]}{dt} = rD[N]([A] - [A_{ex}]) - k_9[A_{ex}] \tag{Eq 6}$$

$$\frac{[dG]}{dt} = k_{10} + \frac{k_4[C]}{k_{DI} + [C]} - k_7[G] - k_{11}[G] \tag{Eq 7}$$

$$\frac{d[N]}{dt} = k_{11}[N]\left(1 - \frac{[N]}{[N_{max}]}\right) \tag{Eq 8}$$

In this study, we numerically solved the system of ODEs using the scipy.integrate.odeint function provided by the SciPy library in Python, which employs LSODA (Livermore Solver for Ordinary Differential Equations). To characterize the "ON" and "OFF" states, simulations were run using Eqs 1–8 with an initial concentration of 1 cell/ml and the variables $I$, $R$, $RA$, $A$, $A_{ex}$, $C$, and $G$ initialized to zero. Within a range spanning low to high maximum achievable cell densities denoted by $N_{max}$, the system exhibits two distinct steady-state solutions contingent upon value of $N_{max}$. These solutions are denoted as the "ON" and "OFF" steady states. Moreover the $N_{max}$ value at which the QS regulated gene, denoted with G, reaches half of its maximum value is defined as the critical cell density required for activation. Throughout the paper, for cells initially in the "OFF" state, the initial conditions in the set of ODEs are set to the values associated with the "OFF" state, while cells initially in the "ON" state are initialized with the values corresponding to the "ON" state. **Fig A in S1 Text** shows expression of the QS target for a range of final cell densities. Molecular concentrations are normalized by dividing by the concentration in the ON state, listed in **Table A in S1 Text.**

The process of dilution resets the external signal concentrations to zero and reduces the initial cell density N to its post-dilution value. When comparing diluted "ON" cells to "OFF" cells, both the external signal concentrations of initially "ON" and "OFF" cells are adjusted to zero.

## Results

### A model for quorum sensing activation and deactivation

Quorum sensing in bacteria is based on synthesis and detection of diffusible signaling molecules known as autoinducers. LuxR-LuxI regulatory circuit in the marine bacterium *Aliivibrio fischeri* is a well-known example of a QS regulatory circuits in Gram-negative bacteria [34]. *A. fisheri* synthesizes the signaling molecule N-3-oxohexanoyl-homoserine lactone (3-oxo-C6-HSL or AHL more generally) using LuxI synthase proteins. Binding to the signal leads to dimerization of the receptor protein, and this dimer acts as a transcription regulator. As the cell density increases, the concentration of AHLs in the environment rises, resulting in an increased number of LuxR dimers and the upregulation of genes regulated by quorum sensing. Notably, the LuxI and LuxR proteins are both upregulated by the receptor dimer, resulting in a positive autoregulatory circuit [35–38]. As a consequence of the positive feedback, activation of quorum sensing resembles a switch-like behavior. This transition to an active state is accompanied by alterations in the levels of several cellular components, including the LuxI and LuxR

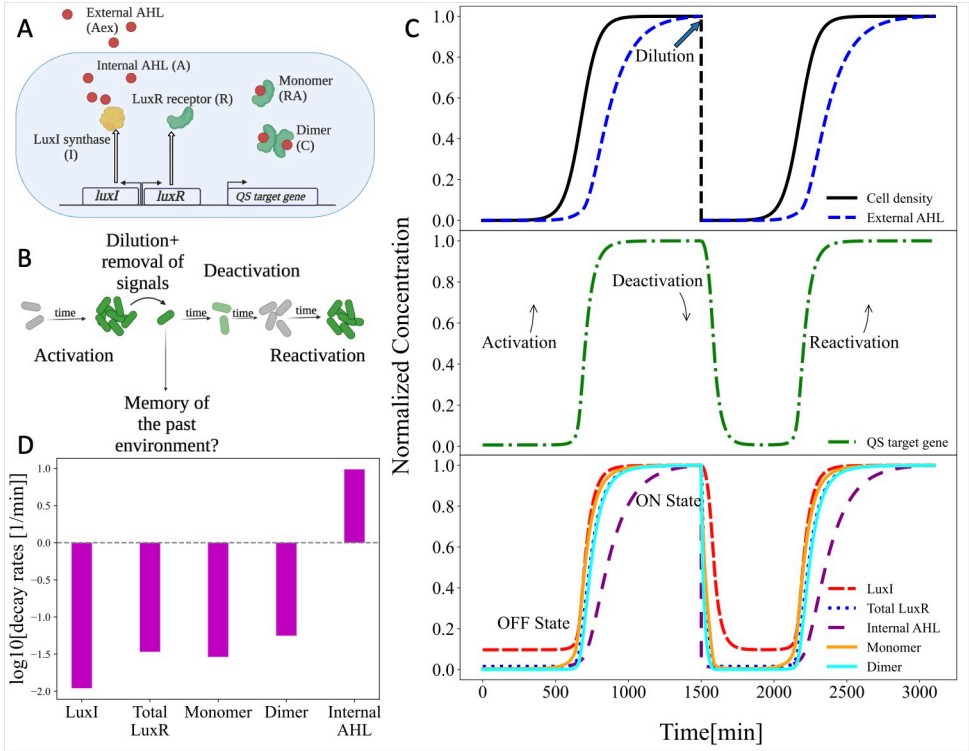

**Fig 1. Dynamics of quorum sensing activation and deactivation. A)** Schematic representation of the LuxR-LuxI QS regulatory circuit. **B)** Cartoon illustration of QS activation and deactivation. Upon dilution of cells to low cell density and removal of extracellular autoinducer, levels of QS-dependent genes begin to decrease. As cells grow and produce signal, QS reactivates. Reactivation is potentially influenced by molecular memory of being in the QS active state. **C)** Simulations examined the concentration of QS-regulated molecules as a population of cells activated and then deactivated QS. As cell density increased, QS-regulated molecules transitioned from low to high concentrations. Eventually cell growth and signal production resulted in the reactivation of QS. **D)** Simulation results were used to calculate the maximum decay rates for each QS-regulated biomolecule during deactivation.

proteins, as well as the concentration of signals [24]. A schematic representation of LuxR-LuxI circuit is shown in **Fig 1A**.

Next, we implemented the model to examine the process of quorum sensing activation and deactivation, as shown in **Fig 1B**. Activation occurs as cells grow and AHLs accumulate over time. Cell density starts at $10^6$ cell/ml and saturates at $10^{11}$ cell/ml. Cells were initialized in the OFF state, using molecular concentrations listed in **Table A in S1 Text**. As shown in **Fig 1C**, the levels of quorum sensing-associated biomolecules undergo a transition from low steady-state concentrations (OFF state) to high steady-state concentrations (ON state).

Although a population of cells held at a high density will remain in the ON state indefinitely, it is possible for cells to transition back to the OFF state under certain conditions, such as if the population is diluted to a very low cell density. To investigate this deactivation process, the model examines a scenario where the cell density is abruptly reduced from $10^{11}$ to $10^6$ cell/ml, and the external AHL concentration is reset to zero. Following the dilution step, the concentrations of quorum sensing-associated biomolecules undergo changes due to several processes: dilution by cell division, intrinsic degradation, transport across the membrane, and dissociation and formation of molecular complexes. Consequently, the concentrations of the QS-associated biomolecules transition from the ON state to the OFF state levels. Moreover, the low concentration of signal in the environment favors loss of signal from inside the cell and therefore a

decrease in the number of dimer complexes, resulting in a reduction in the positive feedback loop that promotes receptor and synthase production in the ON state of quorum sensing. Further cell growth and signal synthesis eventually will lead to QS reactivation. This paper examines the influence of molecular carry-over in initially ON cells on the reactivation dynamics and investigates the conditions in which the phenotypic memory of being in the ON state affects future activity states.

## Carry-over of biomolecules during the deactivation of quorum sensing

In **Fig 1B**, a cartoon illustration depicts the process of removing a cell from a high cell density/high AHL concentration environment. At low cell density, a reduced rate of gene expression along with dilution due to cell division results in a decrease in the concentration of QS-regulated molecules during deactivation. Simultaneously, the reactivation of quorum sensing occurs upon further growth and signal production, which results in increase in concentration of genes under QS regulation.

As seen in **Fig 1C**, upon abrupt changes in cell density and external AHL concentration at $t = 1500$ min, the concentration of QS-regulated genes does not reach OFF-state concentrations instantly, there is a slow decrease in concentrations over a period of time. This transition from ON to OFF state, which occurs mainly due to cell division-mediated dilution, can take multiple generations. Similarly, the normalized concentrations of LuxR, LuxI, monomer, dimers, and the internal AHLs remain elevated for a period of time after the initial dilution. To better understand how multiple QS-related molecules contribute to QS phenotypic memory, the normalized concentration of each biomolecule during deactivation was fitted to an exponential decay equation $C(t) = C_{ON}e^{-\gamma t} + C_{OFF}$. Here C refers to the concentration of each biomolecule, and with $C_{ON}$ and $C_{OFF}$ denoting the characteristic concentrations of these molecules in the ON and OFF states respectively, as listed in Table A in S1 Text. The degradation rate was calculated for LuxR dimers, LuxR monomers, and the total LuxR (the combination of unbound LuxR, bound LuxR, and dimer complex). As shown in **Fig 1D**, The decay rate of internal AHLs is notably high, primarily due to their rapid diffusion across the cell membrane. When external AHLs are removed and cells are diluted to low cell density, the internal AHL concentration decreases to low levels. As a result, the levels of dimers, which are the stable form of LuxR in the ON state [39], decline due to the dissociation into monomers and the degradation of inherently unstable unbound LuxR proteins [25]. This phenomenon leads to a concentration peak of LuxR that persists for approximately two generations. Synthase proteins, LuxI on the other hand, have the slowest decay rates mainly due to dilution by cell division. As seen in **Fig 1C**, the transition from ON to OFF state for LuxI takes up to 4–5 generations. As a result, there are two important biomolecular sources encoding phenotypic memory in quorum sensing, LuxI and LuxR complexes. Due to the carry-over effect of LuxI, cells continue to produce AHLs at a higher rate than the basal level even after being removed from a high cell density environment. This results in a more rapid accumulation of AHLs as compared to signal production from a population of cells in the OFF state. Elevated levels of LuxR presumably make cells more responsive to sense the current AHL concentrations. Next, we explore if this molecular carry-over, QS memory, affects the dynamics of reactivation in a population of cells following dilution to low cell density.

## Phenotypic memory effect in quorum sensing

QS circuits can be approximated as a switch-like activation, requiring a critical signal concentration threshold associated with a critical cell density [40]. This critical density, denoted as $N_C$, is depicted in **Fig A in S1 Text**, in which the expression of QS-regulated genes is shown

for populations grown to different maximum densities. As seen in this figure there is a significant increase in the concentration of QS-regulated genes at approximately $10^{9.8}$ cell/ml. This critical density cell density ($N_C$), is defined as the cell density at which the steady-state concentration of the QS target gene reaches half the maximum value reached in the ON state. We postulated that cells previously in the ON state might reactivate QS at a lower cell density than $N_C$, post dilution to a lower cell density, thereby retaining a memory of past exposure to high signal concentrations.

To test this hypothesis, we compared the activation of QS for cells starting in the ON and OFF states. Cells in initially ON state had the molecular concentration of cells grown to a density of $10^{11}$ cell/ml. Cell in the initially OFF state had the molecular concentration of cells grown to a density of $10^7$ cell/ml. These two populations were simulated with a starting density of $N_i = 10^{8.5}$ and grew to three different final cell densities: $I$: $N_{max} = 10^{8.7}(<N_c)$, $II$: $N_{max} = 10^{9.3}(\approx N_c)$ and $III$:: $N_{max} = 10^{10}(>N_C)$ cell/ml.

As shown in **Fig 2A**, the initial activity state of the cells can determine the final activity state. For densities well below or above $N_C$, the previous activity state of QS had no impact on

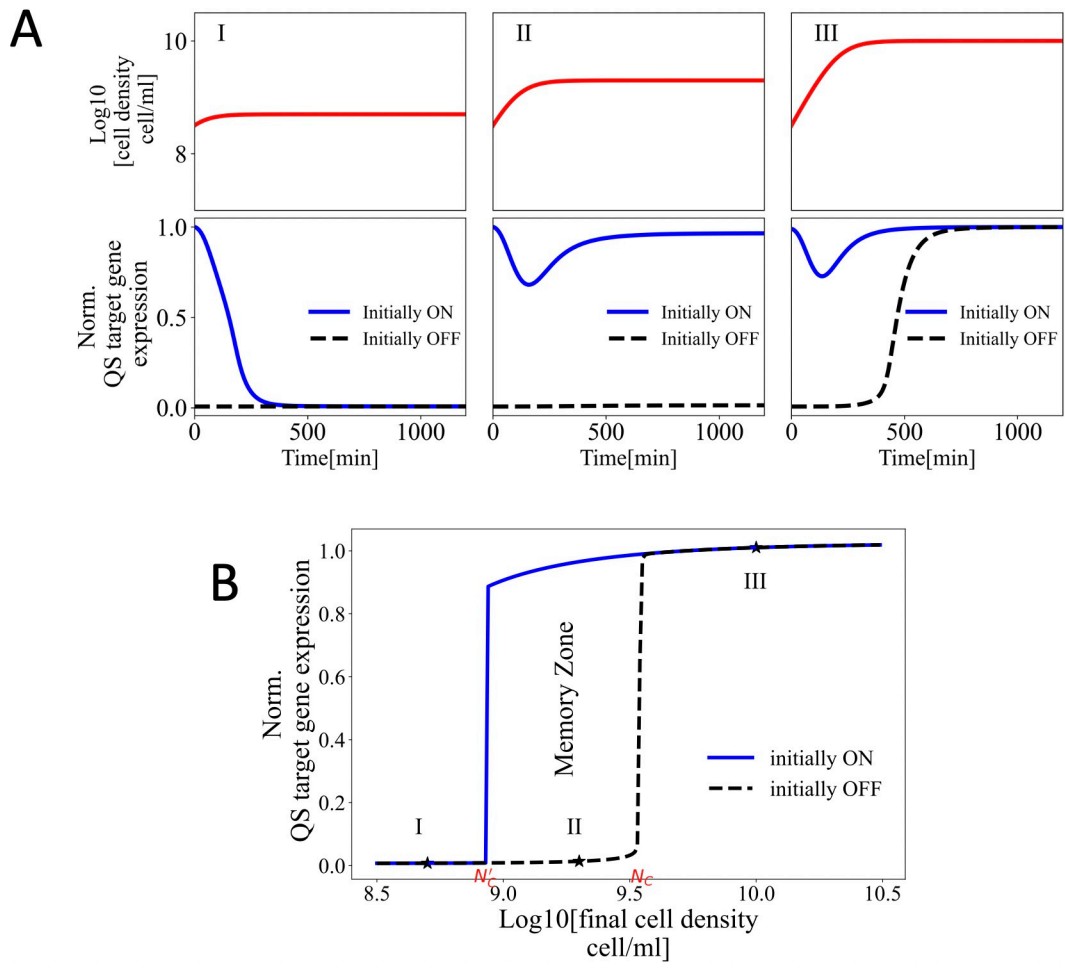

**Fig 2. Phenotypic memory impacts QS reactivation near the critical cell density. A**, top) Populations of cells were grown from a density of $10^{8.5}$ cell/ml to three final cell densities: $I$: $N_{max} = 10^{8.7}<N_c$, $II$: $N_{max} = 10^{9.3}\approx N_c$, and $III$: $N_{max} = 10^{10}>N_c$ cell/ml. **A**, bottom) Normalized QS target gene expression levels for cells initially in the QS ON and QS OFF states, corresponding to each final cell density. **B**) Normalized expression levels of the QS-regulated gene for cells initially in the QS ON and QS OFF states, at the final cell densities in the range of $10^{8.5}$–$10^{10.5}$ cell/ml.

the final activity state. However, for final cell densities near $N_C$, the initial activity state of QS influenced the final activity state. Such history dependent behavior can be seen in **Fig 2A**, when initially ON cells grown to $10^{9.3}$ cell/mL ended up in ON state, while those initially in the QS OFF state remained in the QS OFF state despite reaching the same final cell density. Further simulations of cell growth to a range of final cell densities revealed there exists a range of final cell densities for which the final activity state depended on the initial activity state. As shown in **Fig 2B**, we refer to this range of cell densities as the "memory zone". The range of final cell densities included in this memory zone was dependent on the initial density of the cells following dilution. For example, as shown in **Fig B in S1 Text** over dilution of cells to $10^7$ cell/ml resulted in no memory zone, as in this case the time scale over which the memory of the QS ON state was retained was shorter than the time scale needed for reactivation of QS. Intuitively higher initial cell density leads to accelerated reactivation and the emergence of the memory effect. Conversely, excessive dilution prolongs the time needed for activation and diminishes the effects of the molecular carry-over during the activation process, thereby reducing the extent of the memory zone.

To further explore the effect of each biomolecule in emergence of the memory zone, starting from a high initial cell density, $10^{8.5}$ cell/ml, we switched the value of each component in the initial conditions of the initially ON cells individually to OFF values and plotted the corresponding activation curve as shown in **Fig C in S1 Text.** This figure analyzed the isolated role of different molecular components on QS memory. Switching either dimers or LuxI to OFF values, leads to significant reduction in the memory zone, resulting in an activation curve that overlaps the initially OFF cells. Conversely, the impact of other components, namely unbound LuxR, monomer and the internal AHL concentration on width of memory zone was found to be minimal. This is expected since the dimers are the most abundant form of LuxR in initially ON cells due to stability in presence of AHL. Moreover, LuxI, is expected to encode memory of ON state by maintaining an elevated rate of AHL synthesis hence formation of new dimers and reactivation. These results demonstrate that the most influential phenotypic memory is stored within elevated concentrations of LuxR dimers and LuxI synthase proteins.

Moreover, we explored the reverse scenario by switching the value of each individual component in the initial conditions of the initially OFF cells to ON values and plotted the corresponding activation curve as shown in **Fig C in S1 Text**. As seen in this figure, only when both LuxI and dimer concentrations were switched to ON values the initially OFF cells activated at a lower critical cell density. This result is in agreements with the findings of **Fig C in S1 Text** and confirms that both the synthase and receptor encode the phenotypic memory in QS.

## Cellular parameters controlling the strength of phenotypic memory in quorum sensing

To further explore the other potential parameters influencing phenotypic memory in QS beyond the initial cell density, we developed a simplified analytical approach. This approach predicts the reduction in the critical cell density as a consequence of phenotypic memory.

In this model, we assume that the cells from the QS ON state are diluted to a new cell density, denoted as $N_0$, where $N_0 < N_C$. Upon dilution, external signaling molecules are removed and initially set to a concentration of 0. As seen previously following the dilution, the internal concentration of signals decreases, causing the dissociation of bound receptors, particularly stable dimers, into unbound receptors and free signals. Although the dynamics of signal-receptor dissociation and rebinding depicted in the system of ordinary differential equations are intricate, we make several simplifications. Specifically, we consider that following the removal of signals and dilution, both the dimeric and monomeric configurations of LuxR

molecules rapidly dissociate into individual AHL and LuxR units. This reservoir of receptors, referred to as total LuxR, remains transient before ultimately transitioning to the OFF state. This transient pool of additional LuxR within cells is hypothesized to heighten signal detection sensitivity relative to cells in the OFF state with lower LuxR concentrations. To examine this hypothesis via numerical simulations, we consider the instantaneous disassembly of dimers and monomers upon dilution. As a result, we substituted the LuxR concentration in both dimeric and monomeric forms within initially ON cells with values corresponding to the OFF state, while incorporating the respective values from the instant dissociation into the pool of unbound LuxR and internal AHL concentrations. In **Fig C in S1 Text**, the activation curve for initially ON cells under instant dissociation conditions is shown with a red dashed line. As seen in this figure, the small changes in the activation curves support the assumption of instantaneous dissociation of dimers and signal-bound monomers following the substitution.

Furthermore, presence of this free pool of LuxR, combined with the abundance of synthase proteins (LuxI) in initially ON cells, leads to a potentially accelerated rate of signal- receptor binding events [16], as the formation of complexes between LuxR and signal is a fundamental step for the transition back to the QS ON state. In this section we define activation as the formation of more than a threshold amount of signal bound LuxR.

To begin, we examine the effects resulting from the carry-over of LuxI proteins. Since external signals are removed during cell dilution, we assume that the LuxR dimers dissociate at the time of dilution, decreasing the concentration below the threshold required to sustain elevated production rates of QS-regulated genes. As a result, the rate of gene expression reverts to the basal rate. We assume LuxI is produced at a rate $\beta_I$, equivalent to parameter $k_5$ in **Eq 1** and degrades at a rate of $\gamma_I$. Moreover we assume the decrease in concentration of LuxI after dilution is mainly due to dilution by cell division [41], setting $\gamma_I = \gamma$ to be the cell growth rate which is equivalent to parameter $k_{11}$ in **Eq 1**. Solving **Eq 9**, the changes in LuxI concentration, denoted with I, during the deactivation phase at time $t$ after dilution can be determined as follows:

$$\frac{dI}{dt} = -\gamma I + \beta_I, \qquad\qquad \text{Eq 9}$$

$$I(t) = \frac{\beta_I}{\gamma} + C_I e^{-\gamma t}, \qquad\qquad \text{Eq 10}$$

in which $C_I$ is constant. In the case of complete deactivation, as $t \to \infty$, $I \to I_{\text{OFF}} = \frac{\beta_I}{\gamma}$, where $I_{\text{OFF}}$ represents the LuxI concentration in the OFF state. On the other hand, at $t = 0$, $I_{ON_{t=0}} = \frac{\beta_I}{\gamma} + C_I = I_{OFF} + C_I$. Therefore, since cells are initially in the ON state at $t = 0$, $C_I$ can be calculated as $C_I = I_{\text{ON}} - I_{\text{OFF}}$, where $I_{ON_{t=0}}$ denotes the LuxI concentration in initially ON cells and $I_{\text{ON}}$ denotes the LuxI concentration in the ON steady state. By substituting the value of $C_I$ in **Eq 10**, we can obtain the value of $I(t)$ in initially ON cells as follows:

$$I_{ON_{t=0}}(t) = I_{OFF} + (I_{ON} - I_{OFF})e^{-\gamma t}, \qquad\qquad \text{Eq 11}$$

where $I_{ON_{t=0}}(t)$ denotes the LuxI concentration at time $t$ past dilution initially ON cells. The ratio of the LuxI concentration in the ON state, denoted as $I_{ON}$, to the LuxI concentration in the OFF state, denoted as $I_{OFF}$, is defined as the Fold Change (FC), and can be expressed as $\frac{I_{ON}}{I_{OFF}}$. Fold change in expression of a target gene expressed under QS regulation is a measure of the

promoter activity in active to inactive states. Using this definition, **Eq 11** can be rewritten as:

$$I_{ON_{t=0}}(t) = I_{OFF}(1 + (FC - 1)e^{-\gamma t}).$$

Eq 12

If we assume that each cell synthesizes AHLs at a rate proportional to the concentration of LuxI proteins within the cell, multiplied by a constant rate b, and that the AHLs are immediately released into the extracellular environment, we can calculate the rate of changes in accumulated concentration of autoinducer in the external environment, $A$, using **Eq 13**.

$$\frac{dA}{dt} = v_{cell}N(t)bI(t).$$

Eq 13

This rate can be expressed as the multiplication of the cell volume $v_{cell}$, the cell density $N(t)$, and the AHL synthesis rate $b$, and the concentration of synthase proteins in each cell, denoted as $I(t)$. The cell density $N(t)$ can be described by the exponential growth equation:

$$N(t) = N_0 e^{\gamma t}.$$

Eq 14

where $N_0$ is the initial cell density after dilution, and $\gamma$, represents the growth rate constant. Moreover, we assume that the degradation of AHLs is negligible. By replacing $I$ with the expression in **Eq 12**, we can calculate the accumulated AHL concentration, denoted as $A$, at time t after dilution of initially ON cells:

$$A_{ON_{t=0}} = v_{cell}bI_{OFF} \int_0^t N(t)(1 + (FC - 1)e^{-\gamma t})dt.$$

Eq 15

The integral in **Eq 15** represents the cumulative AHL production over time, and the solution in **Eq 16** provides an explicit expression for the accumulated AHL concentration for initially ON cells as a function of time.

$$A_{ON_{t=0}} = v_{cell}bI_{OFF}N_0 \left( \frac{e^{\gamma t} - 1}{\gamma} + (FC - 1)t \right).$$

Eq 16

**Eq 16** includes two distinct terms. The first term, $\frac{e^{\gamma t}-1}{\gamma}$, characterizes the production of AHL molecules under OFF-level synthase concentration, denoted as $I_{OFF}$, which is applicable to OFF cells. The subsequent term, $(FC-1)t$, is due to the elevated production of AHL molecules as a result of the heightened LuxI presence in initially ON cells. This term is proportional to the initial cell density and quantifies the carry-over effect of heightened levels of LuxI proteins on signal production. **Eq 16** can be rewritten to highlight the contribution of the additional LuxI proteins present at $t = 0$ due to this carry-over effect.

$$A_{ON_{t=0}}(t) = v_{cell}bI_{OFF}N_0 \frac{e^{\gamma t} - 1}{\gamma}(1 + \xi(t)).$$

Eq 17

Here the term $\xi(t)$, here referred to as the "LuxI memory" term, is given by $\xi(t) = \frac{\gamma(FC-1)t}{e^{\gamma t}-1}$. $\xi(t)$ represents the ratio of the accumulated AHLs due to elevated level of LuxI in cells in the ON state over the accumulated AHLs due to basal production, equivalent to the AHL produced by initially OFF cells. For OFF cells, LuxI memory term is always zero, $\xi(t) = 0$.

In **Fig D in S1 Text** the LuxI memory term $\xi(t)$ is depicted with respect to two parameters: the Fold Change $(FC)$, and the doubling time = ln2/γ. Both plots demonstrate that the LuxI memory term is at a maximum at the time of dilution. Subsequently, this value gradually diminishes to zero as time progresses.

Next, we extend this analysis to include carry-over effects resulting from increased concentrations of the pool of LuxR receptor in QS ON cells. By combining the carry-over of LuxR with the increased signal production due to carry-over of LuxI, we can predict the overall contribution to the reformation of LuxR bound with signal.

As shown in **Eq 18**, diluting QS ON cells to a low cell density initiates a slow transition of the concentration of LuxR from $R_{ON}$ to $R_{OFF}$, following:

$$\frac{dR}{dt} = -\gamma_R R + \beta_R, \qquad\qquad \text{Eq 18}$$

$$R_{ON_{t=0}}(t) = \frac{\beta_R}{\gamma_R} + Ce^{-\gamma_R t}, \qquad\qquad \text{Eq 19}$$

Here, the degradation of the total LuxR occurs with rate constant $\gamma_R = \gamma + \gamma_{intrinsic}$, including both dilution by cell division and the rapid intrinsic degradation of LuxR in non-dimeric form.

Solving **Eq 19**, assuming complete deactivation as $t \rightarrow \infty$, $R \rightarrow R_{OFF} = \frac{\beta_R}{\gamma_R}$, where $R_{OFF}$ represents the total LuxR concentration in the OFF state. On the other hand, at $t = 0$, $R_{ON_{t=0}} = \frac{\beta_R}{\gamma_R} + C_R = R_{OFF} + C_R$. Therefore, since cells are initially in the ON state at $t = 0$, $C_R$ can be calculated as $C_R = R_{ON} - R_{OFF}$, where $R_{ON_{t=0}}$ denotes the concentration of total LuxR in initially ON cells at time t, and $R_{ON}$ denotes the concentration of total LuxR in the ON steady state. By substituting the value of $C_R$ in **Eq 19**, we can obtain the value of $R_{ON_{t=0}}(t)$ in initially ON cells as follows:

$$R_{ON_{t=0}}(t) = R_{OFF}(1 + (FC - 1)e^{-\gamma_R t}). \qquad\qquad \text{Eq 20}$$

In **Eq 20**, the concentration of receptors in the OFF state is denoted by $R_{OFF}$, and $R_{ON_{t=0}}(t)$ is the concentration of total receptors (accounting for the dissociation of dimers and monomers) following dilution of QS ON cells at $t = 0$. The Fold Change (*FC*) is assumed to be the same for both LuxI and LuxR and is represented by *FC*.

The rate of formation of bound receptor is determined by a rate constant, here assumed to be equal to one for simplicity, multiplied by the accumulated AHL concentration from **Eq 17**, multiplied by the receptor concentration at time *t* after dilution from **Eq 20**. The formation rate of bound receptors can be written as:

$$R_{ON_{t=0}}(t).A_{ON_{t=0}}(t) = R_{OFF}v_{cell}bI_{OFF}N_0\frac{e^{\gamma t}-1}{\gamma}(1 + (FC-1)e^{-\gamma_R t})(1 + \xi(t)). \qquad \text{Eq 21}$$

**Eq 21** can be rewritten by introducing the overall memory term, $\Theta(t)$, which accounts for the combined carry-over effects of both LuxI and LuxR.

$$R_{ON_{t=0}}(t).A_{ON_{t=0}}(t) = v_{cell}bR_{OFF}I_{OFF}N_0\frac{e^{\gamma t}-1}{\gamma}(1 + \Theta(t)), \qquad\qquad \text{Eq 22}$$

In which,

$$\Theta(t) = \xi(t) + (FC - 1)e^{-\gamma_R t}(1 + \xi(t)). \qquad\qquad \text{Eq 23}$$

$\Theta(t)$ comprises two distinct components. The first term $\xi(t)$ accounts for the carry-over effects of LuxI, whereas the second term accounts for the carry-over effects associated with both LuxI and LuxR. The second introduces a second-order dependency on *FC*. This results in more substantial effects and a quicker decay, attributable to the parameter $\gamma_R$.

**Fig 3A** and **3B** represent the overall memory term $\Theta(t)$, as a function of time past dilution, plotted against Fold Change (*FC*) and doubling time. In **Fig 3A**, doubling time is fixed at 40

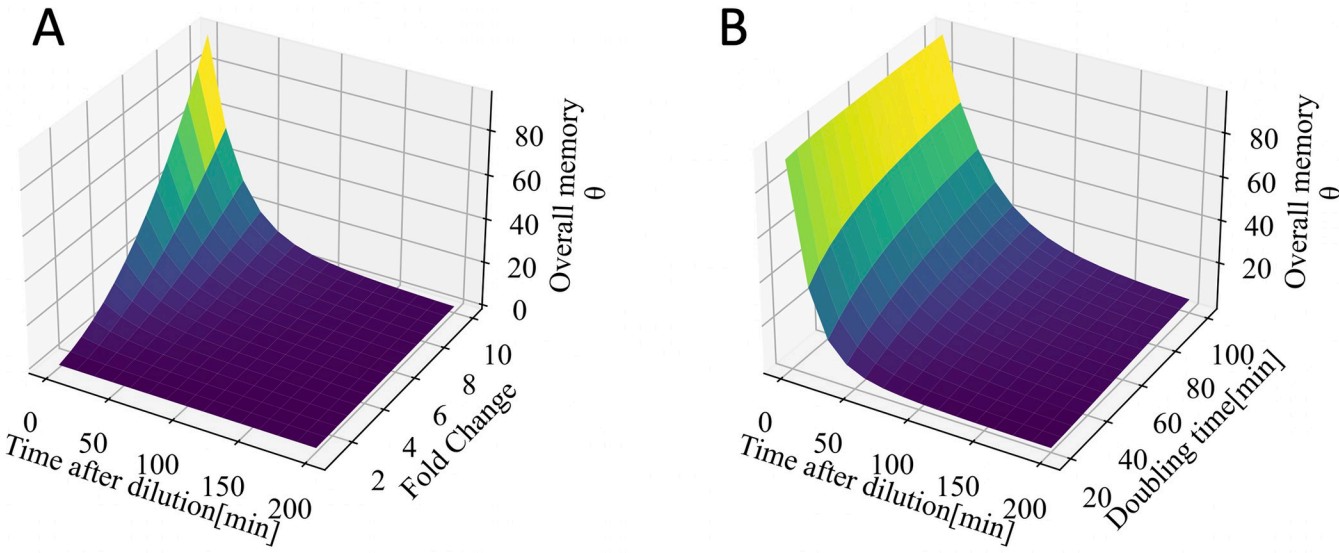

**Fig 3. Overall memory term dependency on time after dilution, Fold Change, and doubling time. A)** Overall memory term, Θ vs time after dilution as a function of Fold Change. Doubling time is set to 40 min. **B)** Overall memory term, Θ vs time after dilution as a function of doubling time. Fold Change set to 10.

minutes, and *FC* values ranges from 1 to 10. It can be observed that the overall memory term dissipates over time, with a more pronounced effect observed for higher *FC* values.

In **Fig 3B**, *FC* is set to 10, and the doubling time varies in range of 20 to 200 min. The overall memory term also dissipates over time, with a more pronounced effect seen for longer doubling times. Intuitively a higher doubling time results in a decrease in the ratio of AHL produced due to the basal rate, compared to the portion synthesized due to carry-over of LuxI in initially ON cells. Larger doubling times lead to a smaller dilution rate of receptors, resulting in a prolonged period of increased sensitivity to AHL concentrations. As shown in **Fig 3B**, the overall memory term is nearly insensitive to changes in the doubling time when the doubling time is large.

We suspect the rapid degradation of LuxR in absence of AHL molecules, decreases the potential memory effect. The figures presented in **Fig E in S1 Text** depict the overall memory term $\Theta(t)$ plotted against Fold Change (*FC*) and doubling time, in the absence of rapid degradation of LuxR ($\gamma_{intrinsic = 0}$, $\gamma_R = \gamma$). Both plots show that in the absence of rapid degradation of LuxR, the overall memory term remains elevated for a longer period. Furthermore, it can be observed from **Fig E in S1 Text** that the overall memory term is more strongly influenced by doubling times in the absence of rapid LuxR degradation, different than the results shown in Fig 3B when rapid LuxR degradation is included. This highlights the impact of fast degradation of LuxR in its non-dimeric form in history-dependent behavior.

## Reactivation of quorum sensing in cells with quorum sensing memory

To further the analysis of QS reactivation following cell dilution, the rate of dimer formation can be calculated by rewriting **Eq 22** in terms of changes in cell density over time, resulting in:

$$R_{ON_{t=0}}(t).A_{ON_{t=0}}(t) = v_{cell}bR_{OFF}I_{OFF}\frac{N(t) - N_0}{\gamma}(1 + \Theta(t)). \qquad \text{Eq 24}$$

To assess the contribution of the overall memory term to reactivation, we assume that once the binding reaction rate in Eq 24, reaches a threshold value, the cells transition instantaneously from the OFF to ON state and remain in a steady ON state indefinitely. We define the activity state as a step function, denoted with $u$, in response to the receptor-signal binding reaction rate as shown in **Eq 24**. We assume when this rate surpasses a certain threshed denoted with $(R.A)_{Thresh}$, the activity state jumps from OFF to ON state here, represented by 0 and 1 states respectively.

$$Activity\ state = u(R(t).A(t) - (R.A)_{Thresh})$$  Eq 25

Using **Eq 24**, the cell density at which reactivation occurs can be calculated. $(R.A)_{Thresh} = R_{OFF}. A_{thresh}$, in which $A_{thresh}$ corresponds to the AHL concentration required for activation. Solving for N, the critical cell density required for activation of initially ON cells, denote by $N'_C$, can be calculated:

$$N'_C = \gamma \frac{A_{thresh}}{v_{cell}bI_{OFF}(1 + \Theta(t))} + N_0.$$  Eq 26

This approach can also calculate the cell density for reactivation of OFF cells, when the value of $\Theta(t)$ is zero. The critical cell density required for activation of initially OFF, $N_C$ is:

$$N_C = \gamma \frac{A_{thresh}}{v_{cell}bI_{off}} + N_0.$$  Eq 27

The memory zone is therefore defined as the range of $N$ values that satisfy the inequality $N'_C < N < N_C$, corresponding to the range of cell densities for which initially OFF cells do not activate, but initially ON cells do reactivate. Where $N_C'$ and $N_C$ are determined using **Eqs 26 and 27**, respectively.

As seen in both **Eqs 26 and 27**, the critical cell density required for activation increases with initial cell density after dilution, $N_0$. Moreover, it has been previously observed when cells are over-diluted with $N_0 << N_C$, activation curves for different dilution ratios overlap [42], leaving the value of the critical cell density unchanged, $N_{C_{N_0 << N_C}} \approx \frac{\gamma A_{thresh}}{v_{cell}bI_{off}}$. This affect is due to the exponential growth within the time interval required for activation, and has been previously observed experimentally [16,42]. By replacing $N_c$ with $N_0 e^{\gamma t_{activation}}$, the time required for activation can be calculated as:

$$t_{activation} \approx \frac{1}{\gamma}\ln\left(\frac{\gamma A_{thresh}}{N_0 v_{cell}bI_{off}}\right).$$  Eq 28

Hence smaller threshold $A_{thresh}$, higher initial cell density $N_0$, and higher AHL synthesis rate constant, $b$, are expected to reduce the time required for activation, regardless of the history of the activity state.

Moreover, higher doubling times always reduce the critical cell density required for activation with a second order dependency on gamma since $I_{off} = \frac{\beta_I}{\gamma}$, $N_{C_{N_0 << N_C}} \approx \frac{\gamma^2 A_{thresh}}{v_{cell}b\ \beta_I}$

The width of the memory zone region can be expressed as $N_C' - N_C$, and can be written as **Eq 29**.

$$N_C - N'_C = \frac{\gamma A_{thresh}}{v_{cell}bI_{off}} \frac{\Theta(t)}{1 + \Theta(t)}|t_{activation_{ON_{t=0}}}.$$  Eq 29

As seen in **Eq 29,** when $\Theta(t)$ is small, the width of the memory zone increases with $\Theta(t)$ at

activation time of initially ON cells, which itself is augmented with higher Fold Change and higher doubling times, and shorter time past dilution. Consequently, a reduced time for activation is anticipated to widen the memory zone region, as the decrease in $\Theta(t)$ over time diminishes the width memory zone.

It's important to note that since $0 \leq \frac{\Theta(t)}{1+\Theta(t)}|t_{activation_{ON_{t=0}}} < 1$, the width of the memory zone always falls between zero and $\frac{\gamma A_{thresh}}{v_{cell} b I_{off}}$. width of zero occurs to the overall memory term at the reactivation time of initially ON cells converges to zero, $\Theta_{t_{act_{ON_{t=0}}}} \rightarrow 0$, resulting in no memory effect, and $N_C' - N_C$. Conversely, an infinitely large overall memory term at reactivation time $(\Theta_{t_{act_{ON_{t=0}}}} \rightarrow \infty)$, leads to $N_C' - N_C$, resulting in the maximum memory zone width of $\frac{\gamma A_{thresh}}{v_{cell} b I_{off}}$.

In **Fig 4A** the QS activity state calculated from **Eq 25**, is shown for initially ON and initially OFF cells as a function of the Fold Change *(FC)* and cell density. The initial cell density in both cases is $N_0 = 10^8$ cell/ml, and cells were allowed to grow exponentially for 700 minutes.

As seen in this figure, for initially ON cells, at higher *FC* values, reactivation occurred at lower cell densities compared to initially OFF cells, resulting in the emergence of the memory zone, shown as purple region. The width of the memory zone varied depending on the *FC* values, narrowing for lower *FC* and widening for larger *FC* values. Such effect is a direct result of higher overall memory term values for higher Fold Changes.

In **Fig 4B,** representing the activity state of initially ON and initially OFF cells, the initial cell density was reduced to $10^6$ cell/ml. Regardless of the value of *FC*, the final activity states of initially ON and initially OFF cells overlap, reducing the width of memory zone to zero. In this case, the delayed reactivation due to over-dilution results in a nearly zero overall memory term at the time of reactivation. This observation aligns with the findings in **Fig B in S1 Text,** where over-dilution results in zero memory effect.

**Fig F in S1 Text** shows the activity state of initially ON and OFF cells as a function of doubling time and cell density. As expected, the width of the memory zone described as **Eq 29** increases with $\Theta(t)$, which, in turn, increases higher doubling time, expands for higher

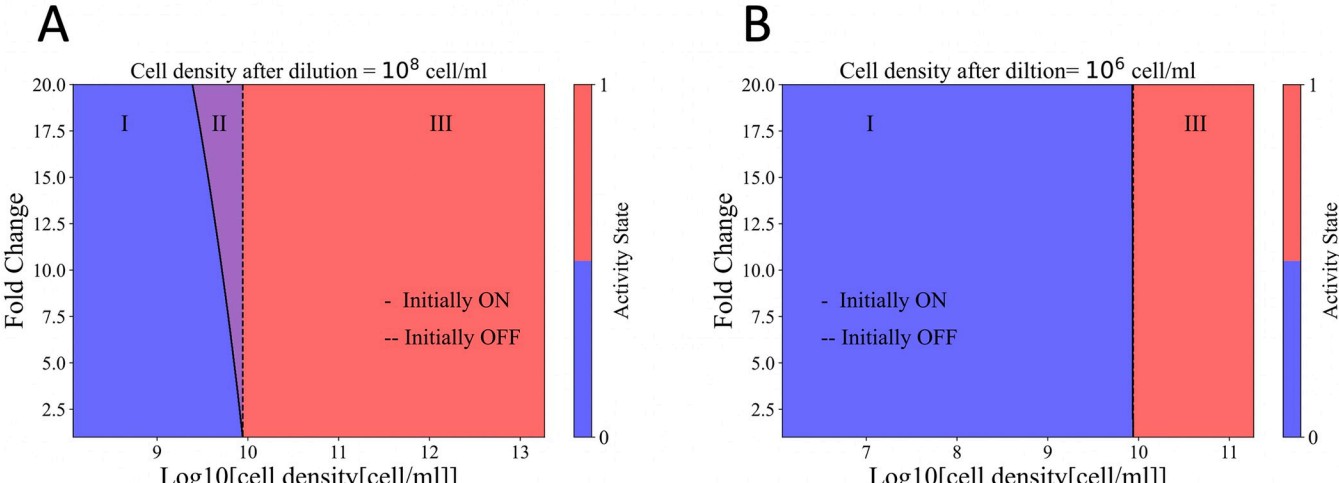

**Fig 4. The effect of Fold Change and initial cell density on the emergence of the memory zone.** The colored regions indicate the state of the cells as density increases. In region I (blue) cells are in the OFF state, in region III (pink) cells are in the ON state, and in region II (purple) the activity state depends on the initial activity state. Solid and dashed lines indicate the boundary between regions I and III for cells in initially OFF or ON states. Cells grow for 700 min, the doubling time is set to 40min, and the Fold Change is kept constant at 10. **A)** The initial cell density is set to $10^8$ cell/ml. Higher *FC* values result in wider memory zone **B)** The initial cell density is reduced to $10^6$ cell/ml, resulting in a delayed reactivation and a decrease in the width of the memory zone to zero. For these calculations $\gamma = 017 min^{-1}$, $b = 0.04 min^{-1}$, $I_{OFF} = 1000 nM$, $A_{thresh} = 20 nM$, $\gamma_R = 0.018 min^{-1}$, $v_{cell} = 10^{-12} ml^{-1}$.

doubling times. Additionally, an increase in doubling time leads to a lower critical cell density required for activation, regardless of the past history of activation.

Moreover, a high basal level of AHL production primes the switch and yields a stronger history-dependent behavior. **Fig G in S1 Text** shows the activity state of initially ON and initially OFF cells per cell density and the AHL basal synthesis rate constant, $b$, which varies between $0.01-0.1 \text{min}^{-1}$. The memory zone widens for higher $b$ and shrinks for smaller values. Moreover, as expected from **Eqs 26 and 27**, higher b reduces the cell density required for activation for both initially ON and initially OFF cells individually, while expanding the memory zone through increasing the overall memory term $\Theta(t)$.

To assess the effect of autoinducer threshold required for activation on memory zone, $A_{thresh}$ values were varied between 10–100 nM. **Fig G in S1 Text** show the activity state of initially ON and initially OFF cells with respect to cell density and $A_{thresh}$. As expected, the cell density required for activation increases with higher $A_{thresh}$ values for both initially ON and initially OFF cells, while the memory zone contracts with increased thresholds. Lower threshold values, like higher basal rate constant values ($b$), lead to earlier reactivation, enhancing the impact of molecular carry-over during reactivation.

## Discussion

Quorum sensing (QS) serves as a regulatory mechanism in bacteria, facilitating coordinated group behavior and optimizing fitness through the controlled production of valuable exoproducts. This mechanism is activated upon reaching a critical cell density [40]. However, the understanding of whether the QS response depends solely on current cell density or is influenced by past activity states, indicating a memory effect, remains limited. This theoretical study aims to investigate the presence of phenotypic memory in QS, specifically focusing on the reduction in critical cell density required for activation as an indicator of memory. Our analysis revealed history of past exposure can affect the future response denoting a memory effect. With the critical cell density require for activation as a measure of phenotypic memory in QS, we showed previously ON cells can activate at a lower critical cell density compared to initially OFF cells.

Activation at lower critical cell densities is largely due to the carry-over of synthase proteins and receptors, which are predominantly in dimeric form in initially ON cells. An excess of synthases in these cells leads to an earlier accumulation of signals above baseline levels compared to initially OFF cells. This increased signal concentration boosts the rate of receptor binding reactions, potentially facilitating the formation of new dimers necessary for reactivation. Moreover, an initial surplus of receptors in ON cells enhances their sensitivity to current signal concentrations by accelerating receptor binding reactions, thus potentially enabling activation at lower cell densities compared to OFF cells. Furthermore, simulation results indicate that both LuxR, mainly in its stable dimeric form, and LuxI play critical roles in sustaining the memory effect by priming the quorum sensing (QS) switch. Substituting either the dimers or LuxI molecules with those from OFF cells results in a significant reduction in the memory effect. Additionally, replacing both LuxI and dimers with ON values in initially OFF cells led to earlier QS activation, emphasizing the role of these components in hastening the QS response. Interestingly, the presence of excess LuxR was found to be a crucial factor for earlier activation, irrespective of its configuration. Substituting dimeric LuxR with unbound forms did not alter the outcomes, suggesting that the availability of LuxR, rather than its dimeric configuration, plays a pivotal role in the activation process. We have shown that this molecular carry-over affects the reactivation specifically near the critical cell density required for activation, where the time required for activation is smaller. We further identified several influential

parameters affecting the strength of the effect. This includes the initial cell density following a change in cell density, the Fold Change in gene expression, the threshold concentration for autoinducers, the basal rate of autoinducer synthesis, and the cell growth rate.

Whether activation at a lower cell density is advantageous or detrimental to a population of bacteria likely depends on the context. For example, previous studies have shown that quorum sensing is the optimal strategy for producing costly public goods when activation occurs at high cell density [43,44]. Consequently, a significant phenotypic memory could hinder the effectiveness of quorum sensing, resulting in the production of public goods at non-optimal cell densities. Another potentially evolved strategy for minimizing significant memory effects in the system could involve non-linear degradation rates of receptors in the absence of autoinducers and the optimization of other cellular parameters, such as the autoinducer threshold and Fold Change.

Although phenotypic memory has been shown to provide advantages in fluctuating environments by optimizing long-term fitness in metabolic regulatory systems [11,45,46], its implications within QS systems remain unclear. Moreover, the optimal strategy for achieving fitness may vary depending on the temporal order and speed of environmental changes. For example, it has been shown that slower or diversified responses is preferred in rapidly changing environments, potentially surpassing strong memory effect [46–50]. The potential disadvantages of QS memory might arise when environmental correlations that historically informed bacterial responses become decoupled due to these rapid environmental changes. In such scenarios, memory-based responses might no longer confer the appropriate adaptive advantages, leading to reduced fitness and possibly increased extinction risks [51]. Additionally, the ability to adapt to changing environments often comes with a fitness cost for growth in stable environments [46].

In natural quorum sensing (QS) systems, fluctuations in cell density, signal concentration, or both, frequently occur. The nonlinear degradation of LuxR in the absence of signals prevents strong memory effects when there is a delay in growth after perturbation. Although LuxI content might not degrade significantly, as dilution by cell division does not occur, delays in growth still lead to degradation processes where receptor dimers dissociate, and unbound receptors degrade. Consequently, by the time favorable growth conditions resume, much of the receptor content may have already degraded, particularly if delays are prolonged, thereby nullifying the memory effect. In quorum sensing (QS) systems, cell density perturbations can occur while external signal concentrations remain unchanged. Enzymes like lactonases and acylases degrade signaling molecules [52], leading to a delayed activation of QS pathways and a reduction in memory effects. These enzymes act as a natural "reset" mechanism, modulating autoinducer levels to prevent the reactivation of QS pathways from residual high-density signals. This adjustment avoids unnecessary energy or virulence factor production and aligns QS responses more dynamically with the current environmental and cell density conditions.

Phenotypic memory in QS is presumably advantageous in periodically fluctuating environment. The disruption of quorum sensing activity through the removal of external autoinducers and reduction of cell density imposes a metabolic cost for re-synthesis of autoinducers [53] until reactivation occurs. In such cases, carry-over effects can be advantageous, enabling cells to initiate a subsequent response at a lower metabolic cost and maximize fitness in the new environment. This memory effect is most prominent when the initial cell density is close to the critical cell density required for activation, ensuring that phenotypic memory is noticeable only when the reduction in cell density is relatively small. Moreover, research has demonstrated that in environments with extreme and random fluctuations, adopting a mixed strategy is often the most effective for survival [15]. Phenotypic memory, in this context, resembles such a mixed strategy, enhancing survivability in unpredictably changing environments.

Ecologically, even transient phenotypic memory can influence interspecies relationships, potentially altering the pattern of associations between species. Additionally, it suggests that the phenotypic memory of species is likely to co-evolve [15]. In environments subject to frequent changes, bacteria equipped with advanced QS memory capabilities can gain a distinct competitive advantage. This allows them to activate survival strategies and dominate ecological niches. Such advantages are especially critical during abrupt environmental changes that demand rapid responses for survival. Consequently, QS memory not only boosts the fitness of individual bacteria but also affects community-level interactions and stability, potentially leading to the development of dominant bacterial strains finely adapted to their fluctuating environments.

Regardless of whether phenotypic memory in QS serves an evolutionary adaptive advantage, it is crucial to study this phenomenon due to various implications in industry and medicine, including QS inhibitors. An example of this is bacteria detaching from a high cell density, QS "ON" environment and infecting small, confined spaces resembling high cell density conditions. This phenomenon is particularly observed in biofilms, where bacteria can disperse from a mature biofilm to colonize new surfaces. The dispersed cells can exhibit behaviors influenced by their previous high cell density environments, potentially accelerating the establishment of new biofilms in confined areas. Phenotypic memory in QS affects population dynamics by enabling faster responses to environmental changes, enhancing competitive advantages in resource utilization and colonization. Moreover, QS memory potentially equips bacteria to adapt more rapidly to environmental fluctuations, which can be crucial for survival in dynamic conditions such as nutrient availability and host immune responses. In pathogenic bacteria, QS memory can modulate the timing and intensity of virulence factor production, potentially affecting their ability to evade or suppress host defenses. Bacteria like *Pseudomonas aeruginosa*, known for its role in chronic infections, often utilize quorum sensing to regulate virulence and biofilm formation, adapting quickly to new environments after dispersing from their original colony. This adaptability is crucial in environments where rapid colonization is necessary for survival, such as in the human lungs during infection. Furthermore, the carry-over effects in QS reactivation can potentially influence the timing and strategy of therapeutic interventions, particularly in QS inhibition approaches. Phenotypic memory could be crucial in strategies for managing infections or treatments, especially considering the frequency and predictability of the environmental pressures involved. The presence of crosstalk and varying thresholds required for the detection to different types of signals [54,55], raises the possibility of a stronger or weaker memory effect in response to fluctuations in specific types of autoinducers, or cell type. This variability can potentially lead to varying degrees of history-dependent behavior, implying potential advantages within complex microbial communities.

It's worth noting that parameters affecting the QS memory exhibit significant variation across bacterial species and environmental conditions [8,33,39,56,57], potentially influencing the strength of the memory effect. For example, the range of Fold Change values in QS circuits can vary significantly, ranging from slightly above one to hundreds. For instance, in *Vibrio harveyi*, LuxR expression demonstrates a Fold Change value of approximately 8 [39]. In contrast, the LasI/LasR system of *Pseudomonas aeruginosa* exhibits a high Fold Change in gene expression, with several hundred-fold increases in LasI and LasR expression at high cell density [56,57]. In this case higher Fold Change facilitate increased production of virulence factors, enhancing competition and colonization of new environments. The optimal Fold Change values in QS circuits likely depend on additional cellular parameters and the fitness costs of differential expression of these genes in response to cell density, rather than solely controlling the carry-over effects. A lower threshold concentration results in activation at a lower critical cell density and a stronger history-dependent behavior upon reactivation. Most bacterial

species exhibit threshold concentrations ranging from 10–50 nM, such as 25-50nM in *Vibrio fisheri* [33]. For example, *Vibrio cholerae* and *Staphylococcus aureus* form biofilms at low auto-inducer thresholds thereby low cell density [8]. Low thresholds are advantageous when rapid responses to changing environmental conditions are needed. On the other hand, higher threshold levels result in delayed reactivation and dampening history-dependent behaviors. Similar to low threshold values, heightened rates of autoinducer production led to faster reactivation, enhancing memory effects. The intricate relationship between these parameters and QS responses in fluctuating environments necessitates further investigation.

Building upon previous experiments demonstrating the presence of phenotypic memory in bacteria [17], we propose to extend these studies to measure the effects of molecular carry-over in cells previously activated ('ON' state) compared to naive ('OFF') cells. Prior work has shown that when a high concentration of signal is added to cells previously in a quorum sensing active state, the carry-over effects led to a faster reactivation of quorum sensing [17]. Future work should further probe quorum sensing memory, testing if carry-over of synthase proteins also changes reactivation dynamics, as predicted in this study. Experimentally probing the limits of QS memory would help determine the specific contexts in which this memory has an impact. To specifically investigate the effects of each biomolecule, the synthase and receptor proteins, genetic circuits could be designed to modulate the Fold Change and degradation rates of each protein independently. To closely mimic the environmental fluctuations that bacterial communities naturally encounter, we recommend exposing bacterial populations to periodically changing signal concentrations and cell densities using microfluidic chambers. Employing microfluidic platforms will enable precise control over experimental conditions [11,58], providing insights into how QS systems adapt over time to dynamic signal environments. The role of cell division and protein degradation in QS memory should also be explored, as these are key factors that set the duration of phenotypic memory. Our findings demonstrated that systems with a high Fold Change, low activation threshold, and high basal levels of signal synthesis exhibit the strongest memory effects, and these results could be experimentally validated. Such studies will not only corroborate our theoretical findings but also expand our understanding of microbial communication in fluctuating environments, potentially leading to novel strategies for managing microbial populations.

It is worth noting that throughout this study the single cell heterogeneity in response was not considered. As a consequence of single cell heterogeneity [59] the population level activation does not necessarily follow a step function, resulting in a graded response [60]. This graded response can, in turn, affect the impact of molecular memory on reactivation. Further study is needed to explore these effects in the emergence of phenotypic memory in quorum sensing.

## Conclusion

In conclusion, investigating the factors influencing phenotypic memory in quorum sensing would provide valuable insights into how bacteria respond to changing environments and optimize their fitness. This knowledge could inform the development of innovative approaches for regulating and manipulating quorum sensing circuits, with potential applications in biotechnology and therapeutics.

## Supporting information

**S1 Text. Supporting figures and table.** S1 Text contains supporting Figs A-G and Table A with concentration values used in simulations.
(DOCX)

**S1 Appendix. S1 Appendix contains a table of data used to generate all main text figures.** (XLSX)

## Author Contributions

**Conceptualization:** Ghazaleh Ostovar, James Q. Boedicker.

**Data curation:** Ghazaleh Ostovar.

**Formal analysis:** Ghazaleh Ostovar.

**Investigation:** Ghazaleh Ostovar.

**Methodology:** Ghazaleh Ostovar.

**Project administration:** James Q. Boedicker.

**Resources:** James Q. Boedicker.

**Software:** Ghazaleh Ostovar.

**Supervision:** James Q. Boedicker.

**Validation:** Ghazaleh Ostovar.

**Visualization:** Ghazaleh Ostovar.

**Writing – original draft:** Ghazaleh Ostovar, James Q. Boedicker.

**Writing – review & editing:** Ghazaleh Ostovar, James Q. Boedicker.

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
