## [Decision Letter · Decision Letter 0]

23 Apr 2024

Dear Dr. Boedicker,

Thank you very much for submitting your manuscript "Phenotypic memory in quorum sensing" for consideration at PLOS Computational Biology. As with all papers reviewed by the journal, your manuscript was reviewed by members of the editorial board and by several independent reviewers. The reviewers appreciated the attention to an important topic. Based on the reviews, we are likely to accept this manuscript for publication, providing that you modify the manuscript according to the review recommendations.

Sincerely,

Oleg A Igoshin

Academic Editor

PLOS Computational Biology

Daniel Beard

Section Editor

PLOS Computational Biology

Reviewer's Responses to Questions

**Comments to the Authors:**

Reviewer #1: The authors use mathematical modeling to examine the conditions favoring phenotypic memory in QS. Using numerical sims and analytical approximations, the authors show that classical QS architectures (positive feedback control of core QS genes) can produce transient memory effects, in particular a heightened initial sensitivity to re-activation following recent de-activation (dilution). The authors further identify the most important cellular currencies and parameters governing this memory phenomenon.

The paper makes a nice logical contribution by showing in a parameterized model that the carry-over of intracellular QS components following dilution can prime the reactivation of QS following subsequent growth. While on some level this ‘has to be true’, the numerical work helps to put some bounds on the importance of this phenomenon, although it has to be said my reading of this is ‘not very important’. Perhaps the authors can push back on this - my reading is that the effects are fairly small, and this is in the favorable case of zero delays (instantaneous dilution, and then immediate entry into favorable growth conditions). It would be good to underline how delays (slower dilution, longer pauses before growth) will rapidly remove all memory effects.

Concerning the importance and impact of this work – I suspect that establishing the importance (or not) of QS memory will require experimental investigation. Adding some words on experimental tests that follow from this work would be helpful.

Methods / Model structure: please state what bug and what environment you are modeling. Table 1 has specific parameters which imply a specific experimental setting defined at a minimum by a species and a medium.

Thinking of signal degradation – many bugs have active QS-controled processes of signal degradation (lactonases, acylases etc) – this might be a design feature to remove memory effects? Possibly something to consider in the discussion?

Line 132: I find the choice of a continuous growth assumption confusing, when elsewhere in the same model you have a saturating logistic growth function. Why not couple growth dilution to your explicit growth equation?

Reviewer #2: The paper "Phenotypic memory in quorum sensing" explores the concept of memory in bacterial quorum sensing (QS) systems. It discusses how bacteria retain information about past environmental conditions through a mechanism known as phenotypic memory, which impacts their future behavior in response to stimuli like cell density and extracellular signal concentrations. Through mathematical modeling and simulations, the authors investigate how various cellular parameters affect the strength and duration of this memory in QS systems. I recommend for its publication if the following concerns could be addressed adequately.

1. The paper provides insights into the transient effects of QS memory but lacks a deep mechanistic explanation of how specific cellular components contribute to this memory.

Suggestion: Expand the discussion on the role of different QS components in memory formation, particularly how proteins like LuxR and LuxI contribute to memory persistence. Include a detailed analysis or additional simulations that isolate the effects of individual components on memory strength and duration.

2. The study is theoretical and heavily relies on simulations. Reference to experimental results and proposals of future experimental validations would be helpful.

3. The conclusion discusses the implications of phenotypic memory in QS but does not fully integrate these findings into the broader context of bacterial behavior and evolution. It would be helpful if the authors could elaborate on how QS memory might influence bacterial population dynamics, virulence, and adaptation in fluctuating environments. Discuss potential evolutionary advantages or disadvantages conferred by phenotypic memory in QS systems.

**Have the authors made all data and (if applicable) computational code underlying the findings in their manuscript fully available?**

Reviewer #1: Yes

Reviewer #2: None

PLOS authors have the option to publish the peer review history of their article (what does this mean?). If published, this will include your full peer review and any attached files.

Reviewer #1: No

Reviewer #2: No

Figure Files:

Data Requirements:

Reproducibility:

References:

---

## [Decision Letter · Decision Letter 1]

19 Jun 2024

Dear Dr. Boedicker,

We are pleased to inform you that your manuscript 'Phenotypic memory in quorum sensing' has been provisionally accepted for publication in PLOS Computational Biology.

Best regards,

Oleg A Igoshin

Academic Editor

PLOS Computational Biology

Daniel Beard

Section Editor

PLOS Computational Biology

---

## [Editor Report · Acceptance letter]

2 Jul 2024

PCOMPBIOL-D-23-01875R1 

Phenotypic memory in quorum sensing

Dear Dr Boedicker,

I am pleased to inform you that your manuscript has been formally accepted for publication in PLOS Computational Biology. Your manuscript is now with our production department and you will be notified of the publication date in due course.

With kind regards,

Zsuzsanna Gémesi
